

# Lithocholic acid induces endoplasmic reticulum stress, autophagy and mitochondrial dysfunction in human prostate cancer cells

Ahmed A. Gafar[1,2], Hossam M. Draz[1,3], Alexander A. Goldberg[1,4], Mohamed A. Bashandy[2], Sayed Bakry[2], Mahmoud A. Khalifa[2], Walid AbuShair[2], Vladimir I. Titorenko[5] and J. Thomas Sanderson[1]

[1] Institut Armand-Frappier, Institut National de la Recherche Scientifique (INRS), Laval, QC, Canada
[2] Zoology Department, Faculty of Science, Al-Azhar University, Cairo, Egypt
[3] Department of Biochemistry, National Research Centre, Dokki, Cairo, Egypt
[4] McGill University Health Centre, Montréal, QC, Canada
[5] Department of Biology, Concordia University, Montréal, QC, Canada

Corresponding author
J. Thomas Sanderson,
thomas.sanderson@iaf.inrs.ca

## ABSTRACT

Lithocholic acid (LCA) is a secondary bile acid that is selectively toxic to human neuroblastoma, breast and prostate cancer cells, whilst sparing normal cells. We previously reported that LCA inhibited cell viability and proliferation and induced apoptosis and necrosis of androgen-dependent LNCaP and androgen-independent PC-3 human prostate cancer cells. In the present study, we investigated the roles of endoplasmic reticulum (ER) stress, autophagy and mitochondrial dysfunction in the toxicity of LCA in PC-3 and autophagy deficient, androgen-independent DU-145 cells. LCA induced ER stress-related proteins, such as CCAAT-enhancer-binding protein homologous protein (CHOP), and the phosphorylation of eukaryotic initiation factor 2-alpha (p-eIF2$\alpha$) and c-Jun N-terminal kinases (p-JNK) in both cancer cell-types. The p53 upregulated modulator of apoptosis (PUMA) and B cell lymphoma-like protein 11 (BIM) levels were decreased at overtly toxic LCA concentrations, although PUMA levels increased at lower LCA concentrations in both cell lines. LCA induced autophagy-related conversion of microtubule-associated proteins 1A/1B light chain 3B (LC3BI–LC3BII), and autophagy-related protein ATG5 in PC-3 cells, but not in autophagy-deficient DU-145 cells. LCA ($>10\,\mu M$) increased levels of reactive oxygen species (ROS) concentration-dependently in PC-3 cells, whereas ROS levels were not affected in DU-145 cells. Salubrinal, an inhibitor of eIF2$\alpha$ dephosphorylation and ER stress, reduced LCA-induced CHOP levels slightly in PC-3, but not DU-145 cells. Salubrinal pre-treatment increased the cytotoxicity of LCA in PC-3 and DU-145 cells and resulted in a statistically significant loss of cell viability at normally non-toxic concentrations of LCA. The late-stage autophagy inhibitor bafilomycin A1 exacerbated LCA toxicity at subtoxic LCA concentrations in PC-3 cells. The antioxidant $\alpha$-tocotrienol strongly inhibited the toxicity of LCA in PC-3 cells, but not in DU-145 cells. Collectively, although LCA induces autophagy and ER stress in PC-3 cells, these processes appear to be initially of protective nature and subsequently consequential to, but not critical for the ROS-mediated mitochondrial dysfunction and cytotoxicity of LCA. The full mechanism

of LCA-induced mitochondrial dysfunction and cytotoxicity in the similarly sensitive DU-145 cells remains to be elucidated.

# INTRODUCTION

Prostate cancer is the second most common cancer worldwide in males and the fourth most common cancer overall, with more than 1,112,000 new cases diagnosed in 2012, representing 15% of male cancer cases and 8% of all cancers (*Ferlay et al., 2015*). In Western men, prostate cancer diagnosis ranks first among male cancers and second as cause of cancer-related death (*Malvezzi et al., 2015*; *American Cancer Society, 2015*; *Canadian Cancer Society's Advisory Committee on Cancer Statistics, 2016*). Standard treatment of prostate cancer consists of surgery (prostatectomy), antihormonal therapy and radiotherapy. Although these treatments are successful for early-stage prostate cancer, they each have potentially serious side-effects (*Martin & D'Amico, 2014*; *Nguyen et al., 2015*), among which some that last a life-time (*Sanda et al., 2008*). Androgen-deprivation therapy uses drugs that blocking the action of male sex hormones either through androgen receptor antagonism (bicalutamide, hydroxyflutamide) or inhibition of androgen biosynthesis (finasteride, abiraterone). These treatments are initially effective in controlling androgen-dependent prostate tumor growth, although side-effects include increased insulin-resistance, bone density loss, hypogonadism, gynecomastia, muscle mass loss and fatigue (*Conde & Aronson, 2003*; *Nguyen et al., 2015*). In addition, a certain percentage of tumors that have undergone androgen-deprivation therapy progresses to an androgen-independent state, which is difficult to treat resulting in increased mortality. The limitations of current standard treatments of prostate cancer has encouraged the search for safer and more effective molecules based on naturally occurring compounds.

Lithocholic acid (LCA) is a secondary bile acid produced by microflora in the gut, which we found to exhibit selective toxicity to human neuroblastoma cells and prostate cancer cells at relatively low concentrations that did not affect normal cells (*Goldberg et al., 2011*; *Goldberg et al., 2013*). LCA triggered both intrinsic and extrinsic pathways of apoptotic cell death that were, at least in part, caspase-dependent. In addition, LCA selectively decreased the viability of human breast cancer and rat glioma cells (*Goldberg et al., 2011*). Various bile acids have been reported to have anti-neoplastic and anti-carcinogenic properties in a number of cancer cell models: chenodeoxycholic acid (CDCA) reduced growth of tamoxifen-resistant breast cancer cells by downregulation of human epidermal growth factor receptor 2 (HER2) promoter activity (*Giordano et al., 2011*), LCA and several of its synthetic enantiomers reduced colon cancer cell proliferation and viability (*Katona et al., 2009*). Deoxycholic acid (DCA), ursodeoxycholic acid (UDCA) and their taurine-derivatives delayed cell cycle progression in Jurkat human T leukemia cells and DCA induced apoptosis (*Fimognari et al., 2009*). These findings indicate that the bile acid
structure may form the basis for the development of potent and selective drugs for the treatment of various cancers including those of the prostate.

The mechanisms underlying the cytotoxicity of LCA are not well understood and remain a continuing topic of investigation. Studies have found that certain bile acids can induce apoptosis via a variety of mechanisms including chronic endoplasmic reticulum (ER) stress (*Perez & Briz, 2009*), autophagy (*Gao et al., 2014*) or disruption of mitochondrial function (*Goldberg et al., 2013*). The endoplasmic reticulum is cell organelle responsible for the synthesis, folding and maturation of proteins, the storage and release of intracellular calcium ($Ca^{2+}$) and a large number of biotransformation reactions. A variety of factors (radiation, pathogens, hypoxia, disease states and chemical agents) can disrupt healthy ER function, resulting in a so-called unfolded protein response (UPR), due to the accumulation of unfolded or misfolded proteins in the lumen of the ER. As an adaptive response to these stress factors, the UPR aims to restore normal cell function by halting protein translation, degrading misfolded proteins and increasing the production of molecular chaperones involved in protein folding. However, chronic activation of the UPR fails to promote cell survival and the cell is broken down by a proapoptotic ER stress-mediated response pathway. CCAAT-enhancer-binding protein homologous protein (CHOP) is a transcriptional regulator induced by ER stress, which is a modulator of ER stress-mediated apoptosis (*Marciniak et al., 2004*) and autophagy (*Shimodaira et al., 2014*). CHOP levels may be increased through activation of various ER stress sensor-pathways, including those initiated by activating transcription factor 6 (ATF6), inositol-requiring enzyme 1 alpha (IRE1$\alpha$) and protein kinase R-like endoplasmic reticulum kinase (PERK), the latter which phosphorylates eukaryotic initiation factor 2-alpha (eIF2$\alpha$), and the downstream transcription factor ATF4 which in turn induces the transcription of CHOP.

Autophagy is a catabolic process for the autophagosomic/lysosomal degradation of bulk cytoplasmic contents (*Reggiori & Klionsky, 2002*; *Codogno & Meijer, 2005*). Autophagy is generally activated by nutrient deprivation but is also important in physiological processes such as fetal development and cell differentiation, as well as diseases such as neurodegeneration, infection and cancer (*Levine & Yuan, 2005*). The molecular machinery of autophagy was largely uncovered in yeast by the discovery of autophagy-related genes (Atg). Formation of the autophagosome involves a ubiquitin-like conjugation system in which Atg12 is covalently bound to Atg5 and targeted to autophagosomal vesicles (*Mizushima et al., 1998a*; *Mizushima et al., 1998b*). Upon induction of autophagy, a fraction of microtubule-associated proteins 1A/1B light chain 3 (LC3-I) is conjugated to phosphotidylethanolamine (PE) to produce LC3-II proteins, which are required for autophagosome membrane expansion and fusion (*Tanida, Ueno & Kominami, 2004*). LC3-I-to-II conversion is reliable marker of autophagosome formation (*Mizushima et al., 2001*).

Bile acids have also been reported to induce apoptosis via disruption of mitochondrial function, ligand-independent activation of death receptor pathways and modulation of certain members of the Bcl2 protein family. We have previously shown that LCA induces intrinsic and extrinsic apoptosis in LNCaP and PC-3 prostate cancer cells that involved a decrease in the mitochondrial protein Bcl-2 and cleavage of Bax, concomitant

with an increase of mitochondrial outer membrane permeability. It has been suggested that the well-known solubilising properties of bile acids could explain disruption of (mitochondrial) membranes and induction of mitochondrial dysfunction leading to cell death. However, the lack of or far poorer toxicity of several enantiomers of toxic bile acids suggests physico-chemical properties alone cannot explain cell toxicity (*Katona et al., 2009*) and that a specific three-dimensional structure is required to explain the selectivity of LCA-mediated toxicity in cancer cells.

Our present study aims to investigate to which extent the involvement of ER stress, autophagy or disruption of mitochondrial function is critical to LCA-induced prostate cancer cell death.

## MATERIALS AND METHODS

### Cell lines and reagents

PC3 and DU-145 cells were obtained from the American Type Culture Collection (Manassas, VA). PC-3 cells were grown in 1:1 (v/v) Dulbecco's modified Eagle medium/Ham's F-12 nutrient mix (DMEM/F12; Life Technologies, Grand Island, NY, USA) supplemented with 10% fetal bovine serum (FBS; Mediatech, Corning, Manassas, VA, USA) and 1% penicillin/streptomycin (Life Technologies). DU-145 and RWPE-1 cells were cultured in RPMI-1640 medium (Life Technologies) supplemented with 10% FBS, 1% HEPES , 1% sodium pyruvate (Sigma-Aldrich, St. Louis, MO, USA) and penicillin/streptomycin. All cells were incubated in a humidified atmosphere of 95% air and 5% $CO_2$ at 37 °C. LCA was purchased from Sigma-Aldrich and dissolved in DMSO as 100 mM a stock solution and 1,000-fold concentrated serial dilutions were prepared in DMSO for treatment of the cells. Bafilomycin A1, salubrinal and D-$\alpha$-tocotrienol (Sigma-Aldrich) were dissolved in DMSO at 1,000-fold stock solutions of 2 $\mu$M, 20 mM and 20 mM, respectively.

### Cell viability

Each cell type was added to 96-well plates at a density of $1 \times 10^4$ cells/well in 200 $\mu$l of complete medium. After 24 h, medium was replaced with fresh medium containing 2% dextran-coated charcoal-treated (stripped) FBS and various concentrations of LCA (0, 5, 10, 25, 50 and 75 $\mu$M) in a final DMSO concentration in culture medium of 0.1%. Cell viability was assessed using a WST-1 Cell Proliferation Reagent kit (Roche, Laval, QC) according to the manufacturer's instructions. Absorbance was measured at 440 nm using a SpectraMax M5 multifunctional spectrophotometer (Molecular Devices, Sunnydale, CA).

### Fluorescence microscopy

PC-3 and DU-145 cells were added to 24-well plates at a density of $1 \times 10^5$ cells/well in 1 ml of complete medium. After 24 h, cells were treated with several concentrations of LCA (0, 1, 3, 10 and 30 $\mu$M) in fresh medium containing 2% stripped FBS and another 24 h later, Hoechst 33342 (Sigma-Aldrich) and propidium iodide (Invitrogen, Carlsbad, CA, USA) were each added at a concentration of 1 $\mu$g/ml per well. After a 15 min incubation

at 37 °C, cells were observed and counted under a Nikon Eclipse (TE-2000U) inverted fluorescence microscope at 20× magnification. Hoechst- and propidium iodide-positive cells were made visible using filter cubes with excitation wavelengths of 330–380 nm and 532–587 nm, respectively. To measure autophagy, PC-3 cells were exposed to LCA (0, 3, 10, 30 and 50 μM) for 24 h and then stained with Hoechst 33342 and 2 μL of Cyto-ID® Green Detection Reagent (ENZ-51031-K200; Enzo Life Science, Farmingdale, NY, USA). After a 15 min incubation at 37 °C, cells were observed and counted under a Nikon Eclipse (TE-2000U) inverted fluorescence microscope at 20× magnification.

## SDS–PAGE and immunoblot analysis

Cells were added to 6-well Cell-Bind plates (Fisher Scientific, Ottawa, ON) at a density of $7.5 \times 10^5$ cells/well in 2 ml of complete culture medium and allowed to adhere for 24 h. Cells were then exposed to LCA (0, 3, 10, 30 and 50 μM) in fresh medium with 2% stripped FBS for 1, 8 or 24 h, dependent on the experiment. Adherent cells were collected using a cell scraper, then rinsed three times in cold phosphate-buffered saline (PBS) followed by centrifugation at 700× g for 5 min. After removing the PBS, the cell pellets were lysed in RIPA buffer containing 1× protease and phosphatase inhibitor cocktail. Then, cell lysates were centrifuged at 15,000 rpm for 15 min at 4 °C and protein concentrations in the supernatant were determined using a BCA protein assay kit (Pierce Biotechnologies, Rockford, IL, USA). Proteins (40 μg) were diluted with loading buffer and boiled for 5 min, then loaded onto 10% sodium dodecyl sulfate-polyacrylamide gels. After electrophoresis, gels were transferred to polyvinylidene diflouride (PVDF) membranes using a Trans-Blot Turbo System (Bio-Rad, Mississauga, ON). Membranes were then blocked using Tris-buffered saline (TBS) containing 5% milk powder (blocking buffer) for 1 h at room temperature, after which the membranes were incubated overnight in blocking buffer with the appropriate primary antibodies (anti CHOP, eIF2α, p- eIF2α, JNK, p-JNK, PUMA, BIM, cleaved caspase 3, LC3BI/II, ATG5 and β-actin at 1:1,000 dilution; Cell Signaling, Beverly, MA) at 4 °C. The next day, membranes were washed three times with Tris-buffered saline containing 0.1% Tween (TBS-T) followed by a 1-h incubation with the appropriate secondary antibody at room temperature. Membranes were washed another three times with TBS-T and then incubated with Immobilon Western Chemiluminescent Horseradish Peroxidase Substrate (EMD Millipore, Billerica, MD, USA) for 5 min to make the bands visible; membranes were sealed in plastic wrap and photographed using a ChemiDoc gel documentation system (Bio-Rad). B-actin was used as reference protein and loading control.

## Gene-silencing using small interfering RNA (siRNA)

CHOP expression was silenced by transfecting PC-3 and DU-145 cells with SMARTpool ON-TARGETplus siRNA oligonucleotides selective for CHOP (Dharmacon, Lafayette, CO) using lipofectamine RNAiMAX (Life Technologies, Burlington, ON, USA) in serum free Opti-MEM according to manufacturer's protocols. ON-TARGETplus Non-targeting Control siRNA was used as negative control. After a 24-h transfection period, cells were exposed to various concentrations of LCA (0, 10 and 30 μM) for 24 h. CHOP protein levels were evaluated by immunoblotting as described above.

## Measurement of reactive oxygen species (ROS)

PC3 and DU-145 cells were added to 96-well plates at a concentration of $1 \times 10^4$ cells/well in 200 μl of their respective culture medium containing 2% stripped FBS. After 24 h, medium was removed and the cells were incubated in prewarmed PBS at 37 °C containing 10 mM fluorescent ROS probe (CM-H2DCFDA; Life Technologies). After 30 min, the PBS mixture was removed and cells were exposed to various concentrations of LCA or 1 μM $H_2O_2$ for 60 min at 37 °C temperature. In experiments with $\alpha$-tocotrienol and N-acetylcysteine, cells were preincubated with the antioxidants for 4 h prior to exposure to LCA. ROS production was quantified using a SpectraMax M5 multifunctional spectrophotometer (Molecular Devices, Sunnydale, CA, USA) with an excitation wavelength of 490 and emission wavelength of 545 nm.

## Statistical analysis

Statistical analyses were performed using GraphPad Prism version 5.0 (GraphPad Software, San Diego, CA, USA). Results are presented as means ± standard deviations of at least three experiments. $IC_{50}$ values were determined from concentration–response curves by non-linear curve-fitting. Statistically significant differences of LCA treatments compared to vehicle control were determined by one-way analysis of variance (ANOVA) and a Dunnett post-hoc test or by two-way ANOVA and a Bonferroni post-hoc test when assessing differences between concentration–response curves. A $p$-value less than 0.05 was considered statistically significant.

# RESULTS

## LCA decreases the viability and induces apoptosis and necrosis of PC-3 and DU-145 human prostate cancer cells

A 24-h exposure to LCA reduced the viability of PC-3 and DU-145 cells concentration-dependently, with $IC_{50}$ values of 32.0 μM and 30.4 μM, respectively (Fig. 1). The viability of RWPE-1 immortalized normal prostate epithelial cells was not affected by concentrations of LCA between 5 and 75 μM (Fig. 1). Hoechst 33342 and propidium iodide-staining of PC-3 and DU-145 cells exposed for 24 h to LCA showed a significant concentration-dependent increase in staining, with both necrotic (and late-apoptotic) and early-apoptotic cells starting to appear at a concentration at or above 3 μM (Fig. 2).

## LCA induces ER stress in PC-3 and DU-145 cells

To determine whether the ER stress pathway was involved in LCA-induced prostate cancer cell death, we determined the concentration- and time-dependent effects of LCA on p-JNK, JNK, p-eIF2$\alpha$, eIF2$\alpha$ and CHOP protein levels, as well as on levels of BIM and PUMA in PC-3 and DU-145 cells exposed for 24 h to sub-cytotoxic (3 and 10 μM) and overtly cytotoxic (30 and 50 μM) concentrations of LCA. Levels of BIM and PUMA were decreased concentration-dependently by LCA in PC-3 and DU-145 cells, although in DU-145 cells PUMA levels increased at 3 and 10 μM before decreasing strongly at overtly cytotoxic concentrations (Fig. 3). LCA concentration-dependently increased levels of p-JNK (46 and 54 KDa) and CHOP (27 kDa) in PC-3 and DU-145 cells (Fig. 3). Phosphorylation of

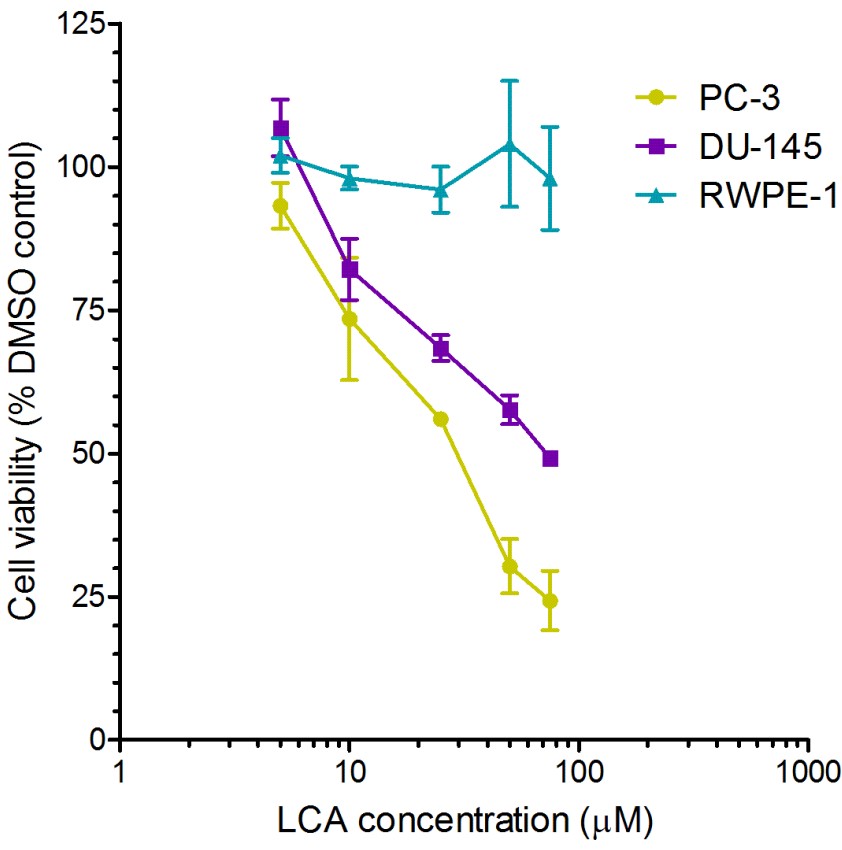

**Figure 1  Lithocholic acid (LCA) decreases the viability of PC-3 and DU-145 human prostate cancer cells, but not RWPE-1 immortalized normal prostate epithelial cells.** Cells were exposed to increasing concentrations of LCA (5–75 $\mu$M) for 24 h. IC$_{50}$ values for LCA-induced cytotoxicity in PC-3 and DU-145 cells were 32.0 $\mu$M and 30.4 $\mu$M, respectively. Experiments were performed three times; per experiment, each concentration was tested in triplicate.

eIF2$\alpha$ was increased in a concentration-dependent manner in DU-145 cells, but was poorly detectable in PC-3 cells after a 24 h exposure to any of the LCA concentrations (Fig. 3).

  To determine the effects of LCA on the ER stress response at earlier time-points, PC-3 and DU-145 cells were exposed to cytotoxic concentrations (30 and 50 $\mu$M) of LCA for 1 and 8 h (Fig. 4). BIM and PUMA levels were decreased concentration-dependently by LCA in both cell lines. In PC-3 cells BIM levels were somewhat higher at 8 h than 1 h (Fig. 4), which appeared to be an effect of the vehicle control, although they were, nevertheless, decreased by LCA, as was observed after 24 h exposure (Fig. 3). In DU-145 cells BIM levels were detectable at 1 h but not at 8 h. PUMA levels were decreased concentration-dependently by LCA in both cell lines, although basal levels in each cell line increased between 1 h and 8 h of culture (Fig. 4). Levels of p-JNK underwent a biphasic response in both cell lines with expression levels appearing lower after 8 h than 1 h of exposure to LCA, whereas levels were increased again after 24 h of exposure, in particular to 50 $\mu$M LCA. Levels of p-eIF2$\alpha$ increased concentration-dependently after a 1 h and 8 h exposure of PC-3 and DU-145 cells to LCA (Fig. 4), but decreased time-dependently

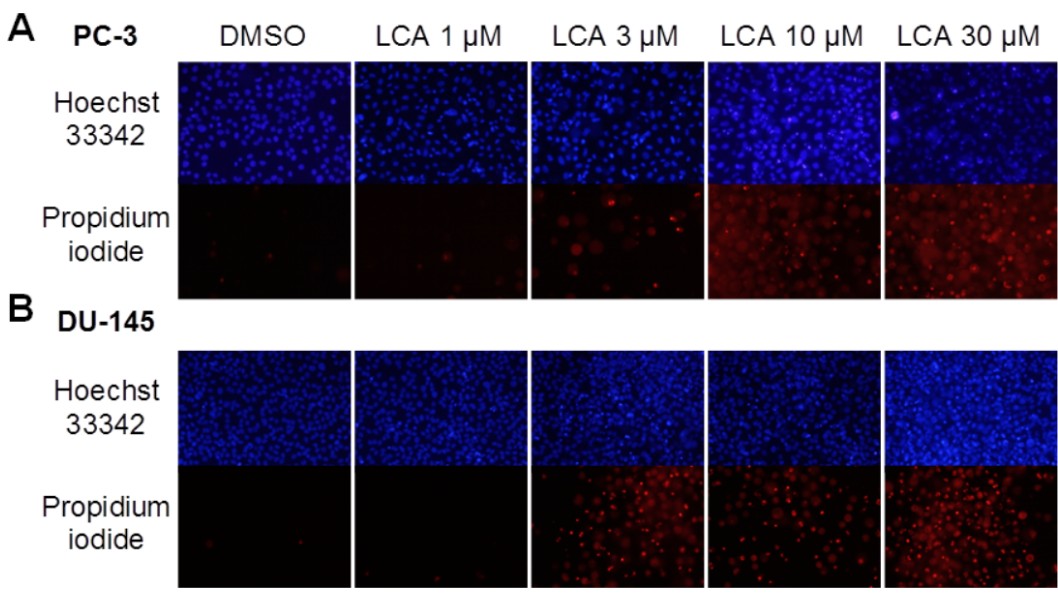

**Figure 2** **Lithocholic acid (LCA) induces apoptotic and necrotic death of PC-3 and DU-145 prostate cancer cells.** Apoptotic nuclear morphology (chromatin condensation) was observed with Hoechst 33342 staining using fluorescence microscopy. Propidium iodide staining was used to distinguish apoptotic from necrotic (and late-apoptotic) cell death. The concentration–response experiment was performed three times using different cell passages. Per experiment, concentrations were tested in triplicate.

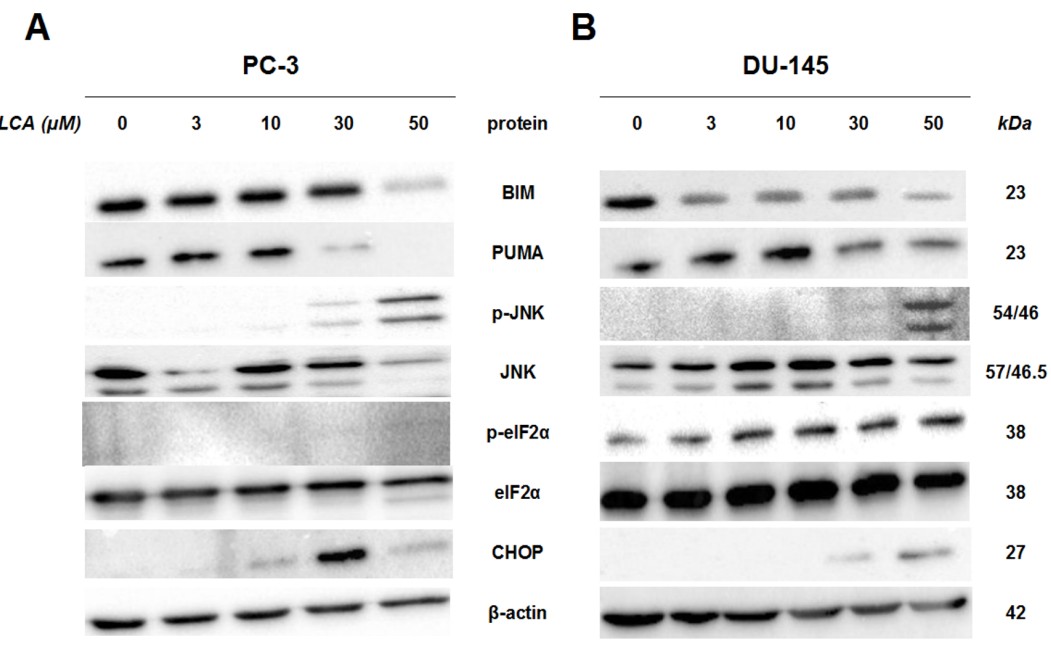

**Figure 3** **Lithocholic acid (LCA) induces ER stress in PC-3 and DU-145 prostate cancer cells.** Cells were exposed to 3, 10, 30 or 50 $\mu$M of LCA for 24 h. BIM, PUMA, p-JNK, JNK, eIF2$\alpha$, p-eIF2$\alpha$, CHOP and $\beta$-actin were detected by immunoblotting; one representative gel of three is shown.

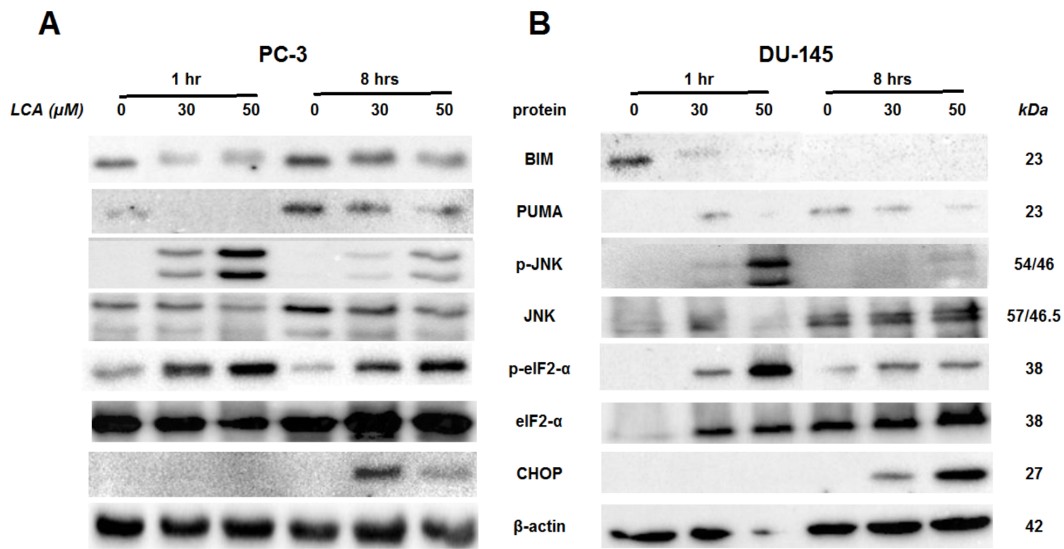

**Figure 4** **Time-dependent induction of ER stress by overtly cytotoxic concentrations of lithocholic acid (LCA) in PC-3 and DU-145 prostate cancer cells.** Cells were exposed to 30 or 50 $\mu$M LCA for 1 and 8 h. BIM, PUMA, p-JNK, JNK, p-eIF2$\alpha$, eIF2$\alpha$, CHOP and $\beta$-actin were detected by immunoblotting; one representative gel of three is shown.

in both cell lines and, after 24 h of exposure, to non-detectable levels in PC-3 cells (Fig. 3). LCA (30 and 50 $\mu$M) visibly increased CHOP levels after 8 h in both cell lines.

## ER stress-inhibitor salubrinal and CHOP gene-silencing do not abrogate LCA-induced cytotoxicity or apoptosis

To determine the role of ER stress in causing the cytotoxicity of LCA to PC-3 and DU-145 cells, each cell type was pretreated for 4 h with salubrinal, a selective inhibitor of eIF2$\alpha$ dephosphorylation, before exposure to toxic concentration of 30 or 50 $\mu$M LCA. After an 8-h exposure, LCA increased levels of cleaved caspase 3, p-eIF2$\alpha$ and CHOP in both cell lines (Fig. 5). Salubrinal pretreatment reduced each of these LCA-mediated increases in PC-3 cells, although in DU-145 cells salubrinal pretreatment increased CHOP levels induced by 50 $\mu$M LCA (Fig. 5). In addition, salubrinal pretreatment did not alleviate LCA-induced death of PC-3 and DU-145 cells, but exacerbated the toxicity of LCA statistically significantly at most test concentrations (Fig. 6).

Given that salubrinal-pretreatment further increased levels of LCA-induced CHOP in DU-145 cells, we assessed the effect of blocking CHOP gene expression using CHOP-selective siRNA. Gene silencing reduced LCA-induced levels of CHOP protein to undetectable levels in DU-145 cells (Fig. 7A). However, no effect of CHOP silencing on LCA-induced cytotoxicity in DU-145 cells was observed (Fig. 7B). This was confirmed using Hoechst staining to evaluate the effect of CHOP silencing on LCA-induced apoptosis in both DU-145 and PC-3 cells (Fig. 8).

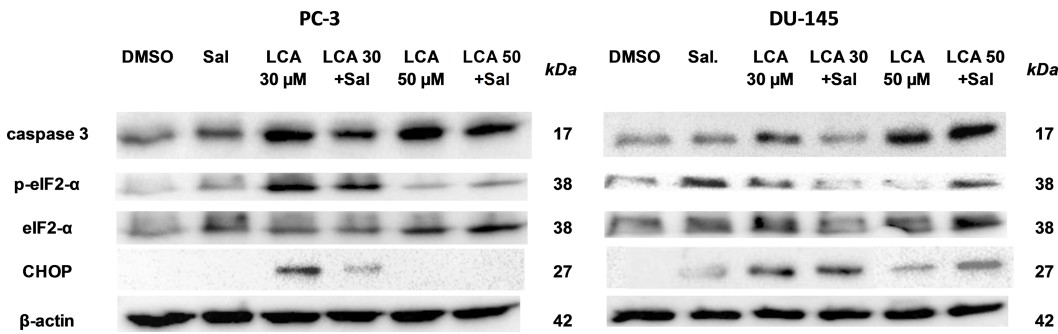

**Figure 5** **The effects of salubrinal-pretreatment on lithocholic acid-(LCA)-induced cleaved caspase 3, p-eIF2α and CHOP levels in PC-3 and DU-145 prostate cancer cells.** PC-3 and DU-145 were exposed to LCA (30 and 50 μM) for 8 h in the presence or absence of 20 μM salubrinal. The expression of caspase-3, p-eIF2α and CHOP was determined by immunoblotting; one representative gel of three is shown.

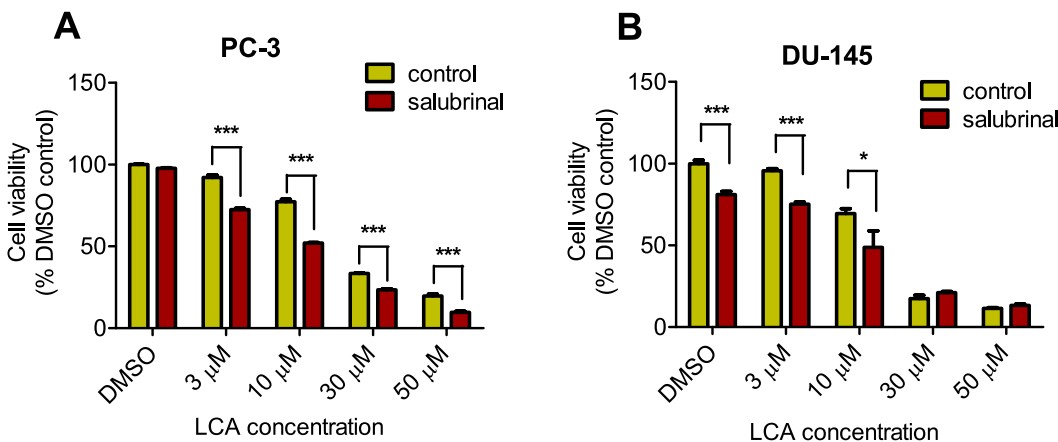

**Figure 6** **Salubrinal-pretreatment exacerbates the cytotoxicity of lithocholic acid (LCA) in PC-3 and DU-145 prostate cancer cells (24 h exposure).** Statistically significant differences in cell viability between salubrinal-treated and vehicle control-treated cells were observed by two-way ANOVA and Bonferroni post-hoc test (* $p < 0.05$; *** $p < 0.001$). Experiments were performed in triplicate using different cell passages; per experiment each concentration was tested in triplicate.

## LCA induces autophagy in PC-3 cells

PC-3 cells exposed to increasing concentrations of LCA for 24 h were stained with Cyto ID Green to detect the formation of autophagic vacuoles. A significant concentration-dependent increase of green fluorescence signal was observed starting at an LCA concentration as low as 1 μM) (Fig. 9). Further confirming the autophagic response, a concentration-dependent increase of the conversion of LC3B I to LC3B II was observed in PC-3 cells (Fig. 10). A time-course experiment indicated that noticeable conversion of LC3B was seen as early as 1 h after exposure to 30 or 50 μM LCA (Fig. 10). When PC-3 cells were pretreated with the autophagy inhibitor bafilomycin A1, the toxicity of relatively non-toxic concentrations of LCA (3 and 10 μM) was increased to a statistically significant degree, whereas no effects on the toxicity of LCA were observed at overtly toxic concentrations of 30 and 50 μM (Fig. 11A). Similarly, silencing LC3B gene expression also

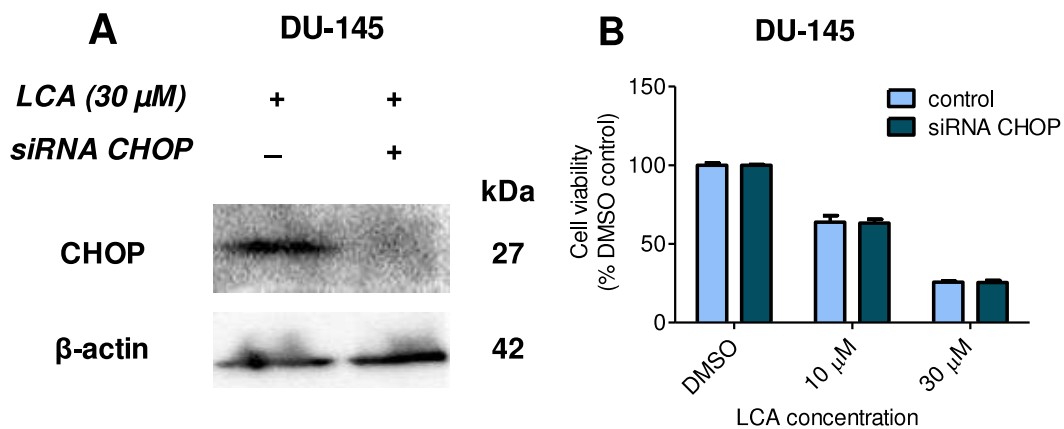

**Figure 7** **CHOP gene silencing does not affect lithocholic acid-(LCA)-induced cytotoxicity in DU-145 prostate cancer cells.** No statistically significant effects were observed of siRNA treatment on control or LCA-decreased DU-145 cell viability by two-way ANOVA ($p = 0.9; n = 3$).

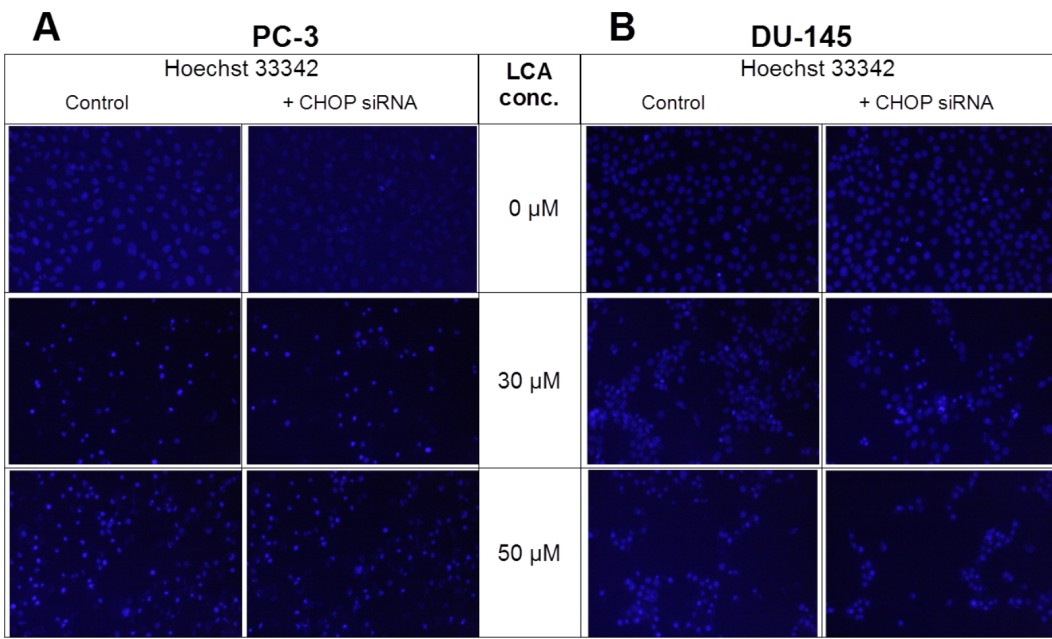

**Figure 8** **CHOP gene silencing does not affect lithocholic acid-(LCA)-induced apoptosis in PC-3 and DU-145 prostate cancer cells.** Apoptotic nuclear morphology (chromatin condensed nuclei) was observed by Hoechst 33342 staining using fluorescence microscopy. The concentration–response experiment was performed three times using different cell passages; per experiment, concentrations were tested in triplicate.

increased the toxicity of LCA at lower concentrations (Fig. 11B). To establish if there was a link between induction of CHOP by LCA and that of autophagy, PC-3 cells were treated with siRNA to silence CHOP and then exposed to 30 or 50 µM LCA (Fig. 12). CHOP silencing did not alter the increased conversion of LC3BI to II or alter the levels of ATG5 protein that were increased by LCA.
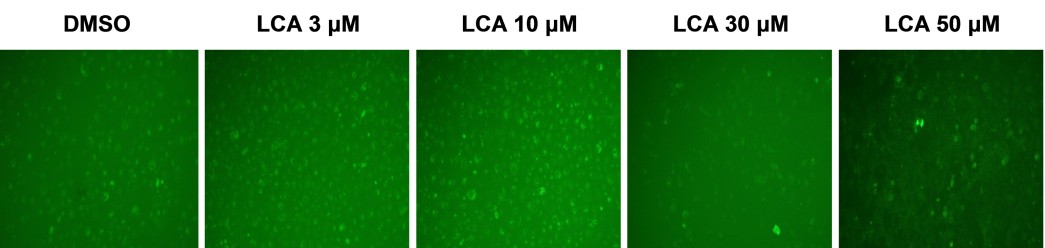

| DMSO | LCA 3 µM | LCA 10 µM | LCA 30 µM | LCA 50 µM |

**Figure 9** **Lithocholic acid (LCA) induces autophagy in PC-3 prostate cancer cells.** Cells were exposed to increasing concentrations of LCA for 24 h and then stained with Cyto-ID® Green dye for 10 min to detect autophagic vacuoles. LCA concentration-dependently increased the accumulation of autophagic vacuoles (bright green fluorescence) as detected by Cyto-ID® Green dye staining using fluorescence microscopy. The concentration–response experiment was performed three times using different cell passages; per experiment, concentrations were tested in triplicate.

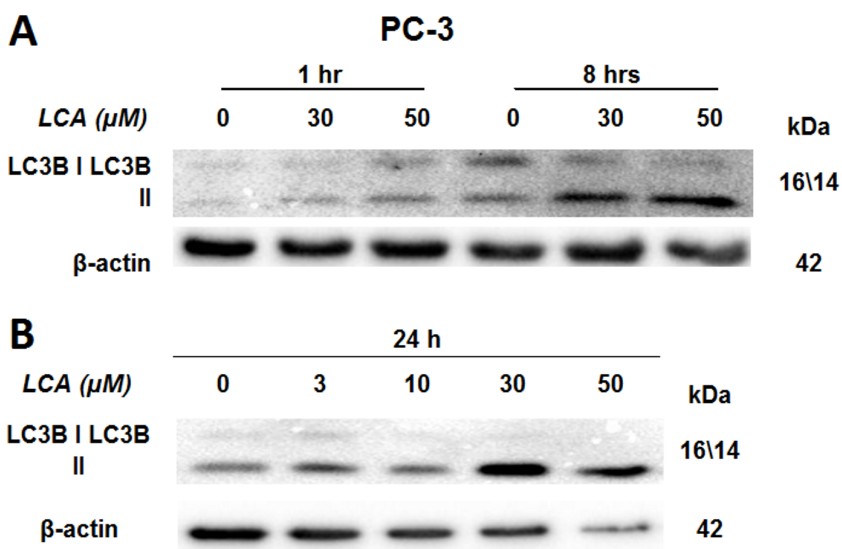

**Figure 10** **Lithocholic acid (LCA) induces LC3B conversion in PC-3 prostate cancer cells.** Cells were exposed to increasing concentrations of LCA for 1, 8 or 24 h. Proteins were detected by immunoblotting; one representative gel of three is shown.

## LCA induces mitochondrial dysfunction in PC-3 and DU-145 cells

Lithocholic acid induced mitochondrial dysfunction in PC-3 and DU-145 as measured using TMRE dye (Fig. 13), which is sequestered by active mitochondria, but fails to accumulate in mitochondria that have reduced or lost their outer membrane potential. PC-3 and DU-145 were exposed to different concentration of LCA (0, 1, 3, 10 and 30 µM) for 8 h and observed a concentration-dependent decrease in TMRE sequestration, which was most apparent at 30 µM LCA (Fig. 13). The loss of mitochondrial membrane potential coincided with an increase in nuclear staining with Hoechst 33342 (Fig. 13).

### LCA induces reactive oxygen species (ROS)

LCA increased the production of ROS concentration dependently in PC-3 but not DU-145 cells at concentrations between 1 and 50 µM. (Fig. 14). To determine if the antioxidant

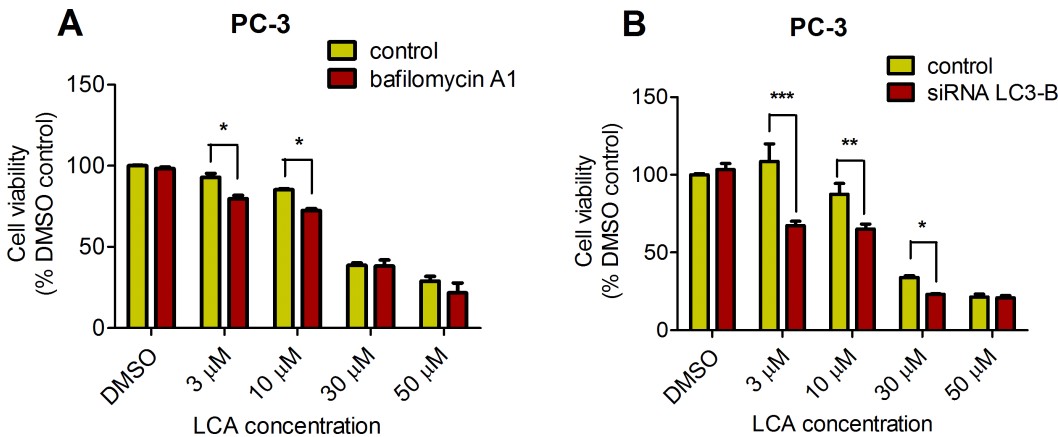

**Figure 11** **Bafilomycin A1-pretreatment (A) or LC3B gene silencing (B) enhanced the cytotoxicity of lithocholic acid (LCA) in PC-3 prostate cancer cells.** Statistically significant differences in cell viability between bafilomycin A1- or LC3B siRNA-treated PC-3 cells and vehicle control-treated cells were determined by two-way ANOVA and Bonferroni post-hoc test (* $p < 0.05$; ** $p < 0.01$; *** $p < 0.001$). Experiments were performed in triplicate using different cell passages; per experiment, each concentration was tested in triplicate.

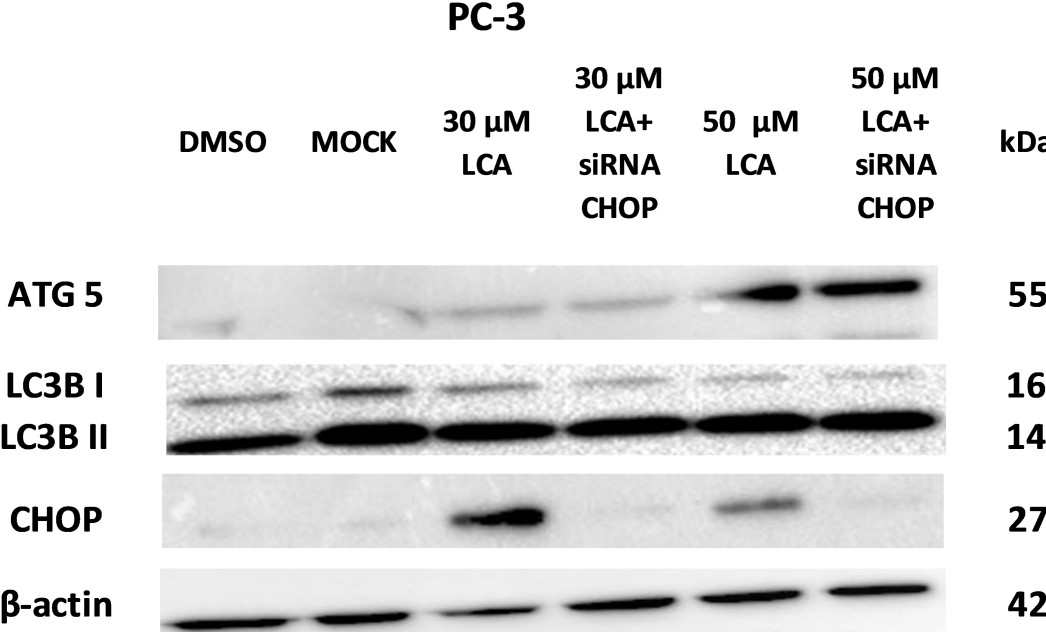

**Figure 12** **CHOP silencing had no affect on LCA-mediated induction of the autophagic markers LC3B conversion or ATG5 expression.** Proteins were detected by immunoblotting; one representative gel of three is shown.

$\alpha$-tocotrienol (T-3; 20 µM) could reduce the cytotoxicity of LCA, PC-3 and DU-145 cells were incubated with T-3 four hours prior to a 24-h exposure to LCA. T-3 protected significantly against LCA-induced cytotoxicity in PC-3 cells whereas in DU-145 cells, T-3 had no effect (Fig. 15).

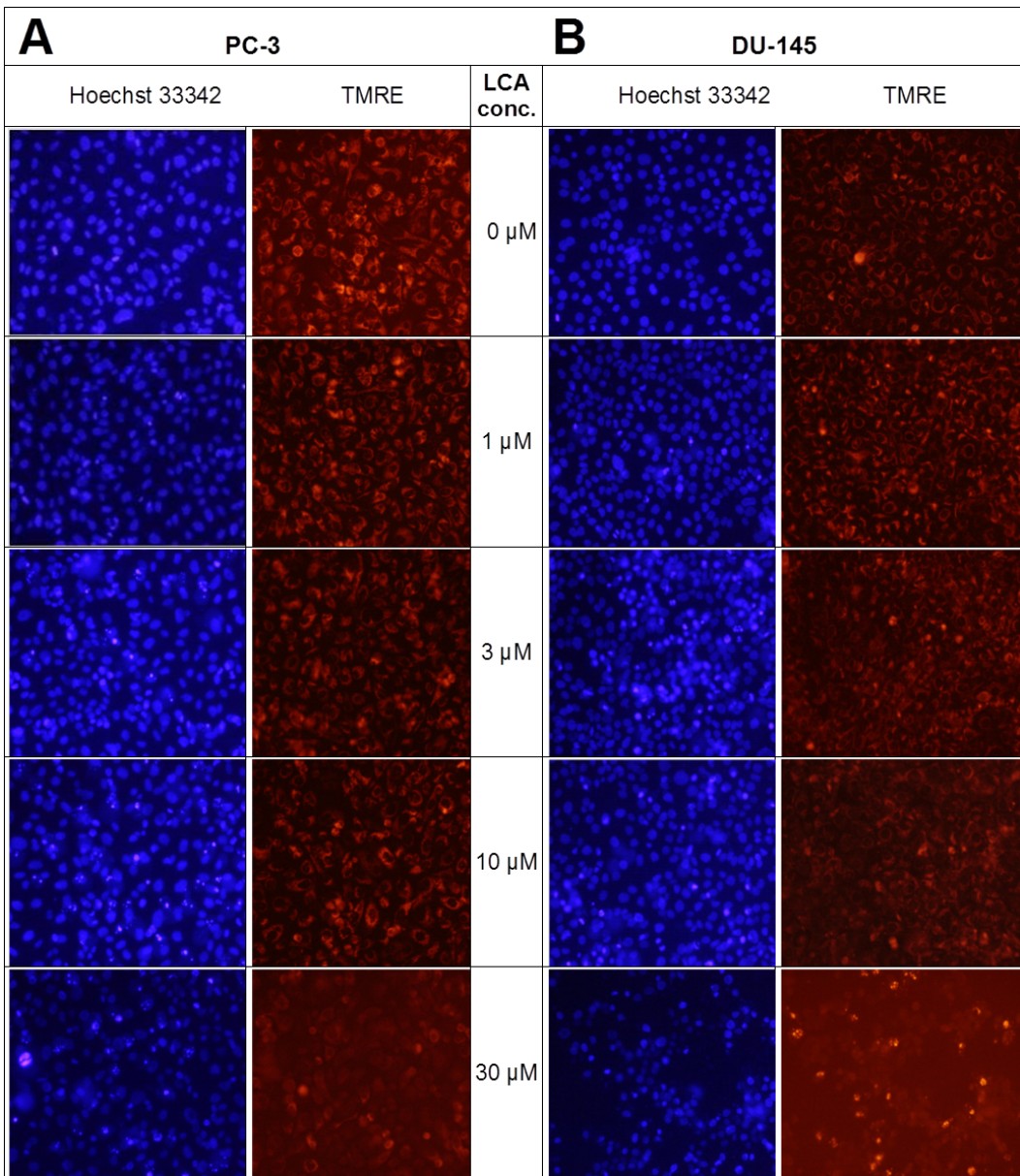

**Figure 13** **Lithocholic acid induces mitochondrial dysfunction in PC-3 and DU-145 cells.** Cells were exposed to different concentrations of LCA for 8 h. Apoptotic nuclear morphology (chromatin condensed nuclei) was observed by Hoechst 33342 staining and mitochondrial membrane permeability was measured using TMRE fluorescent dye by fluorescence microscopy. The concentration–response experiment was performed three times using different cell passages; per experiment, each concentrations was tested in triplicate.

## DISCUSSION

### LCA induces selective cancer cell death

In our study, we found that LCA reduces the viability of androgen-independent DU-145 and PC-3 human prostate cancer cells, but not RWPE-1 immortalized human prostate epithelial cells (Fig. 1), confirming and expanding upon our previous observations in

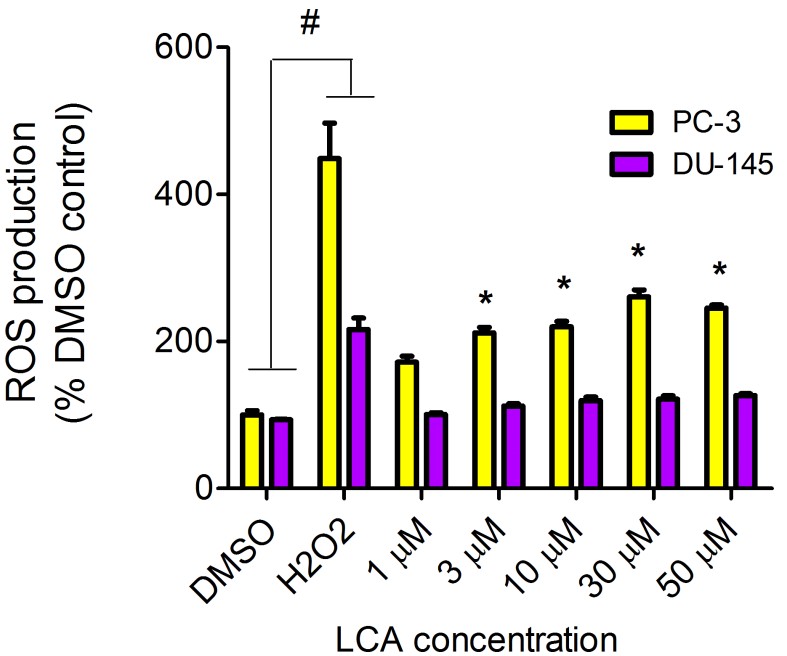

**Figure 14  LCA induces reactive oxygen species (ROS) concentration dependently in PC-3 but not DU-145 cells.** Cells were exposed to LCA for 60 min in culture medium containing 10 mM fluorescent probe dye (CM-H2DCFDA). $H_2O_2$ (20 $\mu$M) was used as a positive control for ROS production, which was measured using a fluorescence spectrophotometer. (#) A statistically significant difference between DMSO- and $H_2O_2$-treated cells. (*) A statistically significant difference between DMSO- and LCA-treated cells determined by one-way ANOVA followed by a Dunnett test. One of three experiments is shown; each concentration was tested in triplicate.

prostate cancer cells that included androgen-dependent LNCaP cells (*Goldberg et al., 2013*). LCA triggered concentration-dependent death of PC3 and DU-145 cells via apoptotic and necrotic pathways (Fig. 2). The selectiveness of LCA in killing cancer cells has recently been demonstrated in hepatocytes, where galactosylated poly(ethylene glycol)-conjugated LCA was toxic to HepG2 human hepatocarcinoma cells, but not to immortalized human LO2 liver cells (*Gankhuyag et al., 2015*). Furthermore, we have previously shown that LCA killed neuroblastoma cells, whilst sparing normal neuronal cells (*Goldberg et al., 2011*).

## LCA induces ER stress in prostate cancer cells

We show for the first time that LCA induces ER stress in human androgen-independent prostate cancer cells in a time- and concentration-dependent manner (Figs. 3 and 4). Toxic concentrations of LCA reduced BIM and PUMA, and increased CHOP levels and the phosphorylation of eIF2$\alpha$ and JNK in both cancer cell types. Increased phosphorylation of eIF2$\alpha$ and JNK were early (1 h) responses to toxic concentrations of LCA, whereas concentration-dependent decreases of BIM and PUMA were sustained between 1 and 24 h of exposure (Figs. 3 and 4). The increased cleavage of caspase 3 by LCA (Fig. 5) likely explains why BIM and PUMA levels decreased at toxic concentrations of LCA, as it is known that active caspase 3 downregulates PUMA (*Hadji et al., 2010*) and BIM (*Wakeyama et al., 2007*) expression in other cell types. At lower LCA concentrations

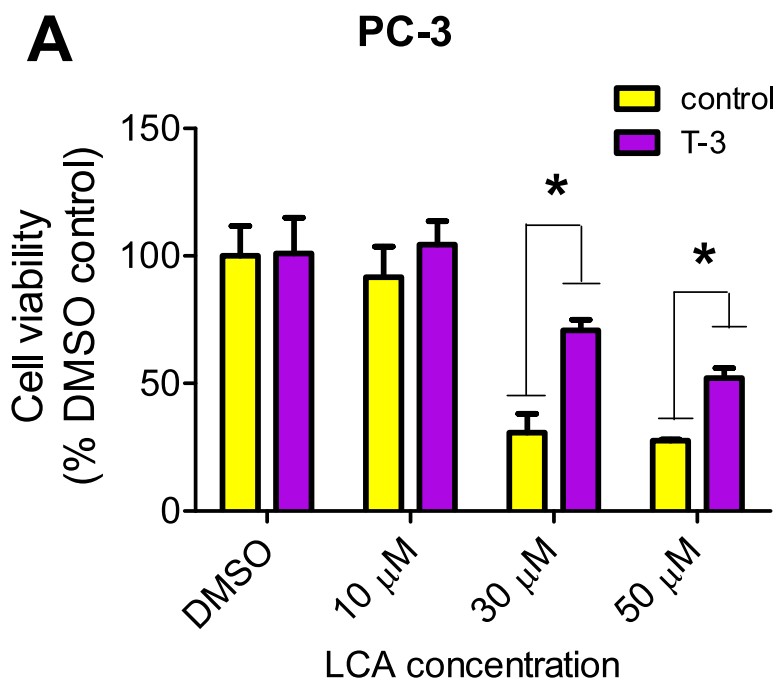

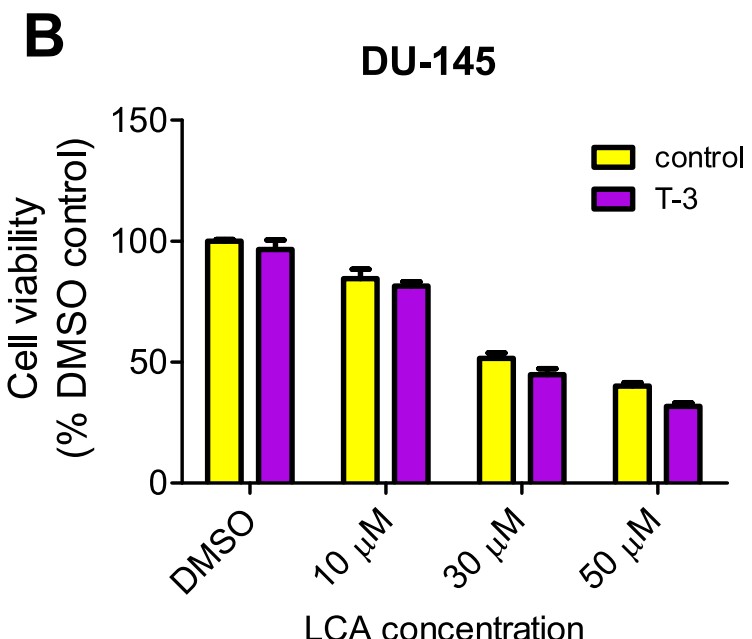

**Figure 15** **Effects of a 4-hour pretreatment with the antioxidant α-tocotrienol (T-3; 20 μM) on the cytotoxicity of LCA (24 h exposure) in (PC-3 cells or DU-145 cells.** Statistically significant differences in cell viability between antioxidant-treated and vehicle control-treated cells were observed by two-way ANOVA and Bonferroni post-hoc test (* $p < 0.05$). One of three experiments is shown; each concentration was tested in triplicate.

and at earlier exposure durations, on the other hand, PUMA is initially increased, suggesting that PUMA is involved in triggering mitochondrial apoptosis (as discuss later) and caspase 3 activation that ultimately results in its breakdown. The up-regulation of PUMA is clearly p53-independent in PC-3 cells as these cells are p53-deficient (*Rubin et al., 1991*).

LCA caused sustained induction of CHOP at 30 μM in PC-3 cells, although levels were sharply lower at 50 μM, possible due to excessive cell death (Figs. 3–5). In DU-145 cells CHOP levels were increased by 30 and 50 μM LCA, but levels declined between 8 and 24 h of exposure (Figs. 3 and 4). Our observations suggest that LCA-induced ER stress involves activation of the eIF2α phosphorylation pathway and subsequent induction of p-JNK (early response) and CHOP (later response), resulting in caspase 3-dependent apoptosis. However, an attempt to block this particular pathway with salubrinal reduced CHOP induction in PC-3 cells only, although it decreased LCA-induced caspase 3 in both cell lines (Fig. 5). Yet, salubrinal pretreatment resulted in increased toxicity of LCA in both cell lines (Fig. 6). We have previously shown that direct inhibition of the catalytic activity of caspase 3 did result in partial protection against LCA-induced cytotoxicity in LNCaP and PC-3 prostate cancer cells (*Goldberg et al., 2013*), and in neuroblastoma cells (*Goldberg et al., 2011*). It is possible that the observed decreases in cleaved caspase 3 protein levels do not reflect a significant change in its catalytic activity.

Furthermore, blocking CHOP expression using *CHOP*-selective siRNA had no effect on the reduced viability (Fig. 7) or apoptosis (as determined by measuring chromatin condensation and fragmentation using the fluorescent dye Hoechst 33342) (Fig. 8) of DU-145 and PC-3 cells after exposure to increasing concentrations of LCA). Therefore, inhibition of ER stress signaling alone does not appear to be essential for LCA-induced prostate cancer cell death.

Other studies have observed the induction of ER stress by bile acids. In HepG2 cells, the secondary bile acids LCA and DCA were the most toxic, followed by CDCA, although they induced cell death at concentrations of 100 μM and above (*Adachi et al., 2014*), which are significantly greater than the concentrations of LCA that we have found to be toxic to prostate cancer cells. The same investigators detected increased expression of genes involved in ER stress, such as *GRP78* and *CHOP* after 24 h exposures to 100 μM of LCA, DCA or CDCA. Using CDCA as a prototype bile acid, it was found to increase caspase 3 activity at 200 μM, but not 100 μM. Although cytotoxicity and CHOP induction, but not caspase 3 activation, appeared to occur concurrently after exposure to certain bile acids (*Adachi et al., 2014*), a direct link between ER stress and HepG2 cell death was not established. Glycochenodeoxycholic acid (GCDCA) has been shown to induce ER stress in freshly isolated rat hepatocytes and this study interestingly showed that ER stress-mediated activation of caspase 12 occurred at a later stage than mitochondrial apoptosis mediated by cytochrome c release and caspase 3 activation (*Tsuchiya et al., 2006*), suggesting induction of ER stress may not be critical to cell death. In a follow-up study, the investigators determined that caspase 8 activation via the extrinsic Fas pathway triggered ER stress in response to 300 μM GCDCA in HepG2 human hepatocarcinoma cells (*Iizaka et al., 2007*). It is unclear how critical caspase activation is for bile acid-induced cell death. Glycodeoxycholate induced caspase 3-dependent apoptosis in rat hepatocytes after a 2 h

exposure and inhibition of caspase 3 activity resulted in less apoptosis, but whether this translated into less cell death was not reported (*Webster, Usechak & Anwer, 2002*). We point out that these previous studies were performed with remarkably high concentrations of bile acids and whether cells were dying due to excess necrosis was never reported. We have previously shown in LNCaP and PC-3 prostate cancer cells that LCA (50 and 75 μM, respectively) activates caspases 8, 9 and 3, and that caspase 9 activation was likely secondary to caspase 8-induced truncation of Bid (*Goldberg et al., 2013*), a finding consistent with those of *Iizaka et al. (2007)*. Inhibition of caspases 8 or 3 resulted in partial protection against LCA induced cytotoxicity, suggesting that the cytotoxicity of LCA is, at least in part, caspase-dependent (*Goldberg et al., 2013*). However, we are currently performing studies to show that necrotic signaling pathways may play a significant role in LCA-induced death of prostate cancer cells.

## LCA induces autophagy in PC-3 cells

We found that LCA induces a general autophagic response in PC-3 cells based on a time- and concentration-dependent increase of LC3B conversion observed in these cells (Figs. 9 and 10). To delineate the protective or cytotoxic nature of the autophagic response to LCA, cells were exposed to LCA after pre-incubation with the autophagy inhibitor bafilomycin A1. Cells were also treated with siRNA specific for LC3B to silence the expresssion of this protein. Inhibiting autophagy in PC-3 cells in either of these manners enhanced the toxicity of normally sub-cytotoxic concentrations of LCA (Fig. 11A and 11B). This observation indicates that the autophagic response of PC-3 cells to LCA exposure is, at least initially, of a protective nature. Similarly, autophagy was shown to provide protection against cell death of rat hepatocytes induced by glycochenodeoxycholate, as its inhibition using the autophagy inhibitor chloroquine exacerbated toxicity whereas induction of autophagy using rapamycin provided protection against cell death (*Gao et al., 2014*). Our laboratory has also recently shown that blocking autophagy in LNCaP and LNCaP C4-2B prostate cancer cells, resulted in a strong sensitization of these cells to the cytotoxicity of diindolylmethane and a series of ring-substituted dihalogenated DIM derivatives again demonstrating the protective nature of the autophagic process in these cells (*Goldberg et al., 2015*).

To our knowledge, this is the first reported observation that LCA induces autophagy in human (prostate) cancer cells, although a link between bile acids and autophagy has been recently proposed via activation of the farnesoid X receptor (FXR) (*Nie, Hu & Yan, 2015*). The FXR is a cytoplasmic receptor and an important target for hydrophilic primary bile acids, but is unlikely to play a large role in the biological effects of LCA, which is very hydrophobic and remains almost entirely outside the cell (*Goldberg et al., 2013*). More likely targets for LCA are cell membrane surface receptors such as the death receptors or the G-protein-coupled bile acid receptor (GPBAR1), the latter for which LCA has a particularly strong affinity. Although the role of the GPBAR1 in LCA mediated signaling in healthy cells is currently under intense investigation (*Tiwari & Maiti, 2009*; *Stepanov, Stankov & Mikov, 2013*; *Fiorucci & Distrutti, 2015*; *Li & Chiang, 2015*; *Perino & Schoonjans, 2015*), nothing is known about its functions in prostate cancer cells. Our preliminary results

show strong expression of GPBAR1 protein in LNCaP, PC-3 and DU-145 cells and we are currently investigating the role of this receptor in triggering various cell death or survival pathways in these prostate cancer cells.

We did not establish a link between the induction of ER stress by LCA and its induction of autophagy. CHOP silencing did not alter the autophagic response of PC-3 cells to LCA at the tested concentration of 30 and 50 µM as we observed no changes in the induction of LCB3 conversion or ATG5 protein levels (Fig. 12). A recent study showed that whether triggering ER stress resulted in induction of either autophagy or apoptosis depended on the type of trigger. They found that triggering ER stress with thapsigargin only resulted in induction of apoptosis, whereas the ER stress inducer tunicamycin only caused autophagy (*Matsumoto et al., 2013*). However, it was not made clear whether the induction of either autophagy or apoptosis was directly mediated by ER stress or could have been due to off-target effects of the typical ER stress inducers. Our results indicate that the induction of ER stress by LCA was not directly responsible for the induction of either cell death or autophagy, and that likely these effects are secondary to the disruption of mitochondrial function by LCA.

## LCA induces mitochondrial dysfunction in PC-3 and DU-145 cells

We have shown that LCA impairs mitochondrial function by increasing mitochondrial outer-membrane permeability (Fig. 13). These results confirm our earlier finding that LCA impairs mitochondrial membrane potential in PC-3 and LNCaP cells as early as 1 h after exposure and was sustained for at least 8 h (*Goldberg et al., 2013*). In the present study, we found that induction of ROS by LCA (Fig. 14) appeared to be a key trigger of cell death in PC-3 cells as the antioxidant T-3 was able to protect these cells against the cytotoxicity of LCA (Fig. 15). Interestingly LCA did not induce ROS in DU-145 cells (Fig. 14) and consistent with this, antioxidant pretreatment had no protective effect against LCA-mediated cytotoxicity in these cells (Fig. 15). These remarkable differences in (anti)oxidative responses between the two cell lines warrant further investigation.

## CONCLUSIONS

In summary, we have found that LCA induces an ER stress response in PC-3 and DU-145 human prostate cancer cells via a p-eIF2$\alpha$-dependent pathway and an autophagic response in autophagy-capable PC-3 cells. These pathways appear to play a cytoprotective role against LCA-induced cell death, and are rather a response to the underlying, yet to be precisely elucidated mechanisms of LCA-induced prostate cancer cell death. These underlying mechanisms appear to involve induction of ROS and subsequent mitochondrial dysfunction in PC-3 cells, whereas in DU-145 cells LCA-induced mitochondrial dysfunction and cell death occurred at similar LCA concentrations, yet in the absence of ROS formation.

### Funding
This study was supported financially by an Egyptian government scholarship from the Administration of Cultural and General Affairs and Missions to Ahmed Gafar and by the Canadian Institutes of Health Research (CIHR) of Canada (grant no. MOP-115019) to Thomas Sanderson. Hossam Draz was funded by a scholarship from the Fondation Universitaire Armand-Frappier INRS. The funders had no role in study design, data collection and analysis, decision to publish, or preparation of the manuscript.

### Grant Disclosures
The following grant information was disclosed by the authors:
Administration of Cultural and General Affairs and Missions to Ahmed Gafar.
Canadian Institutes of Health Research (CIHR) of Canada: MOP-115019.
Fondation Universitaire Armand-Frappier INRS.

### Competing Interests
Thomas Sanderson is an Academic Editor for PeerJ, but was not involved in the review process of this manuscript.

### Author Contributions
- Ahmed A. Gafar conceived and designed the experiments, performed the experiments, analyzed the data, wrote the paper, prepared figures and/or tables.
- Hossam M. Draz performed the experiments, analyzed the data, reviewed drafts of the paper.
- Alexander A. Goldberg analyzed the data, reviewed drafts of the paper.
- Mohamed A. Bashandy, Sayed Bakry, Mahmoud A. Khalifa and Walid AbuShair contributed reagents/materials/analysis tools, secured Egyptian funding for Ahmed Gafar's stipend and research materials.
- Vladimir I. Titorenko reviewed drafts of the paper.
- J. Thomas Sanderson conceived and designed the experiments, analyzed the data, contributed reagents/materials/analysis tools, wrote the paper, prepared figures and/or tables, reviewed drafts of the paper.

### Data Availability
The raw data for Figs. 1, 6, 7B, 11, 14 and 15 were supplied as a Data S1.

### Supplemental Information
Supplemental information for this article can be found online at http://dx.doi.org/10.7717/peerj.2445#supplemental-information.

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
