# Peer review of "Lithocholic acid induces endoplasmic reticulum stress, autophagy and mitochondrial dysfunction in human prostate cancer cells"

_PeerJ, doi:10.7717/peerj.2445_

## Round 0.1 · original submission · Major Revisions

This is an interesting study examining the effects of LCA on prostate cancer cell death and the effect on ER stress and autophagy. Although the experiments were straightforward, the reviewers raised several important questions which need to be addressed to improve the manuscript.

·

Basic reporting

No Comments

Experimental design

No Comments

Validity of the findings

No Comments

Additional comments

I would suggest a major revision before publication. My specific comments are as follows.
1. The whole manuscript should be revised for some typewriting errors.
2. In Figure 1, the unit of LCA is missing.
3. The character of apoptosis in Figure 2 is not clear. More details should be provided.
4. The number and company of Cyto-ID®Green dye should be added in the  Materials and Methods section. And the Cyto-ID®Green dye measures autophagic vacuoles and monitors autophagic flux in live cells, but the image in Fig.8 was not shown it.
5. Some more experiments should be performed for better understanding of the 
relationship between ER stress and autophagy.

Reviewer 2 ·

Basic reporting

The authors have studied the effect of lithocholic acid on ER stress and autophagy in human prostate cancer cells. Data are clearly presented and manuscript is relatively well written. However, there are some major flaws in the experiment rationale and data interpretation, which greatly weakens this study.
1. It is an interesting finding that lithocholic acid at <75 uM can cause cell death of prostate cancer cells but not normal prostate epithelial cells. In the line of depicting the mechanism underlying the different cell viability, cytotoxic pathway should be studied. However, autophagy itself is generally considered as a cytoprotective and cell survival pathway. ER stress has recently been shown to either stimulate or inhibit autophagy.
2. There are theories about the mechanism for cytotoxicity of bile acids. Mitochondrial permeability transition is thought to be important in mediating bile acid cytotoxicity. And bile acids at low concentrations (< 100 uM) are generally thought to cause apoptosis, rather than necrosis. There may be cell-specific response and mechanism involved, however, this should be at least well discussed in the manuscript to give the big picture.
3. More background information is needed on ER stress and autophagy in the introduction.

Experimental design

1. The study design is mainly descriptive, and lacks of a logic mechanistic connection between cell death, ER stress, and autophagy.
2. The unfolded protein response which is closely related to ER stress is overlooked.
3. The rationale for measuring various proteins (CHOP, eIF2a, JNK, PUMA, BIM, LC3B, etc) is not clearly illustrated. The full names of these proteins should be spelled out the first time they are mentioned in the manuscript and the functions of these markers should be well explained to aid in the data interpretation of readers.

Validity of the findings

1. Fig.2 figure legend: "Propidium iodide staining was used to distinguish apoptotic from necrotic cell death" is inappropriate. PI staining is not a good marker, because both late apoptotic and necrotic cells are PI positive. Annexin V/PI staining followed by flow cytometry should be performed.
2. The data description of western blotting of ER stress-related proteins is not accurate (Fig. 3 and Fig. 4). For example, Page 8, line 176-179, the changes of protein levels should not be called "concentration-dependent increase/decrease", because there are several exceptions. The authors should be more careful when describing the data as well as the following data interpretation in the discussion.
3. The responses in the two androgen-independent human prostate cancer cells DU-145 and PC-3 are not entirely same. The authors are encouraged to provide explanations for this discrepancy.

Minor comments:
1. Page 2, line 25: "loxic" should be "toxic"
2. Page 3, line 59: "various bile acids", please give examples
3. Page 4, line 74-76, Page 5, line 96: the company information needs to be provided regarding the source where reagents and chemicals are derived.

Reviewer 3 ·

Basic reporting

This manuscript provides evidence that LCA treatment of prostate cancer cells caused ER stress and autophagy, which appear to exert protective effects against LCA-induced cell death. Such findings, if solidified, will help better understand the mechanism of LCA-induced death of cancer cells.

Experimental design

Chemical inhibitors of eIF2α phosphatase and autophagy were used to study the role of ER stress and autophagy in LCA-induced death of prostate cancer cells. Chemical inhibitors often have off-target effects. This manuscript will be strengthened if other approaches (e.g. siRNA knockdown) or multiple inhibitors of each pathway were used.

Pretreatment with Salubrinal enhanced the toxicity of LCA to prostate cancer cells (Fig. 6). It would be interesting, and potentially therapeutically important, to determine whether such co-treatment will enhance the toxicity of LCA to normal prostate cells.

Validity of the findings

As expected, Salubrinal treatment increased p-eIF2-α in both cell lines. However, cotreatment with Salubrinal and LCA caused varying results on the p-eIF2-α levels in the two cell lines. Because only single samples were used in the Western blot without normalization, it is hard to judge the reliability of Western blot data and the data interpretation. Multiple samples from each group should be used with statistics to verify some key Western blot data.

It is not clear whether the various experiments have been repeated. In particular, there is discrepancy in the induction of CHOP after LCA treatment. In Fig. 5, the induction of CHOP by LCA was lost at 50 uM after 8 h treatment in both cell lines, whereas in Fig. 4, induction of CHOP by LCA peaked at 50 uM after 8 h treatment.

Additional comments

The English of this manuscript needs improvement.

---

## Round 0.2 · Minor Revisions

The authors took 6 months to substantially revise this manuscript by performing siRNA experiments with CHOP and LC3B, by further examining mitochondrial function studies with more images, western blots, and the use of antioxidant α-tocotrienol. The authors have successfully answered all the questions and this manuscript is in a form acceptable for publication in PeerJ pending some remaining minor revisions.

In addition, the paper could be condensed to be more concise and to the topic. For example, Abstract is now 329 words, and the abstract limit in PubMed is 250 words. In ASPET Journals, the words limit for Introduction is 750 (this paper is over 1000), and for Discussion is 1500 (this paper is now 2070). Although PeerJ does not have such limits, some condensation could be desired. The Figures were increased from original 10 to 15, with many additional additions. Some figures could be used as Supplemental figures (such as Figure 13) to make it more concise.

Reviewer 2 ·

Basic reporting

see below

Experimental design

see below

Validity of the findings

see below

Additional comments

The authors have successfully addressed most of the questions, and only a few more writing errors remain to be corrected.
Line 84: the authors need to cite more reference for the statement “bile acids can induce apoptosis via a variety of mechanisms including chronic ER stress (Ref), autophagy (Ref) or disruption of mitochondrial function (Ref).”
Line 82: found certain bile acids to induce-change to “found that certain bile acids can induce”
Line 87: health-change to “healthy”
Line 361: withan-change to “with an”
Line 432: may be-change to “is possible”
Line 459: dying of-change to “dying due to”
Line 500: studied-change to “study”
Line 501: dependended-change to “depended”.
Line 502: triggering of-change to “triggering”
Line 504: the induction of-change to “the induction of”
Line 506: either-change to “the induction of either”

---

## Round 0.3 · accepted · Accept

This is a nice and solid story. "The good story is simple and to the point". A concise abstract could be better.